# Cx43 Hemichannel and Panx1 Channel Modulation by Gap19 and ^10^Panx1 Peptides

**DOI:** 10.3390/ijms241411612

**Published:** 2023-07-18

**Authors:** Alessio Lissoni, Siyu Tao, Rosalie Allewaert, Katja Witschas, Luc Leybaert

**Affiliations:** Department of Basic and Applied Medical Sciences—Physiology Group, Ghent University, 9000 Ghent, Belgium; alessio.lissoni@ugent.be (A.L.); siyu.tao@ugent.be (S.T.); rosalie.allewaert@ugent.be (R.A.)

**Keywords:** Cx43, Panx1, channel gating, single-channel analysis

## Abstract

Cx43 hemichannels (HCs) and Panx1 channels are two genetically distant protein families. Despite the lack of sequence homology, Cx43 and Panx1 channels have been the subject of debate due to their overlapping expression and the fact that both channels present similarities in terms of their membrane topology and electrical properties. Using the mimetic peptides Gap19 and ^10^Panx1, this study aimed to investigate the cross-effects of these peptides on Cx43 HCs and Panx1 channels. The single-channel current activity from stably expressing HeLa-Cx43 and C6-Panx1 cells was recorded using patch-clamp experiments in whole-cell voltage-clamp mode, demonstrating 214 pS and 68 pS average unitary conductances for the respective channels. Gap19 was applied intracellularly while ^10^Panx1 was applied extracellularly at different concentrations (100, 200 and 500 μM) and the average nominal open probability (NP_o_) was determined for each testing condition. A concentration of 100 µM Gap19 more than halved the NP_o_ of Cx43 HCs, while 200 µM ^10^Panx1 was necessary to obtain a half-maximal NP_o_ reduction in the Panx1 channels. Gap19 started to significantly inhibit the Panx1 channels at 500 µM, reducing the NP_o_ by 26% while reducing the NP_o_ of the Cx43 HCs by 84%. In contrast ^10^Panx1 significantly reduced the NP_o_ of the Cx43 HCs by 37% at 100 µM and by 83% at 200 µM, a concentration that caused the half-maximal inhibition of the Panx1 channels. These results demonstrate that ^10^Panx1 inhibits Cx43 HCs over the 100–500 µM concentration range while 500 µM intracellular Gap19 is necessary to observe some inhibition of Panx1 channels.

## 1. Introduction

Connexins and pannexins are two ubiquitous but distinct protein families forming membrane hemichannels (HCs) and channels in the plasma membrane, respectively. They facilitate the exchange of ions and small molecules (MW ≤ 1.5 kDa) between the intracellular and the extracellular compartment upon opening. Among these respective families, connexin-43 (Cx43) and pannexin-1 (Panx1) are the most abundant isotypes in the human body expressed in a wide array of tissues. Panx1 and Cx43 have several common properties, including the following: (i) an analogous tetra-spanning transmembrane topology with cytoplasmic N- and C-termini (NT and CT), two cysteine-containing extracellular loops (ELs) and one cytosolic loop (CL); (ii) a large diameter channel pore in the order of 17–21 Å; (iii) poor ion selectivity; and (iv) channel opening in response to electrical, mechanical and intracellular calcium ([Ca^2+^]_i_) elevation. The opening of Panx1 channels and Cx43 HCs is triggered by positive membrane voltages [1,2]. However, Panx1 calcium dependency is currently a matter of debate, since there is also evidence of the channel not being sensitive to moderate [Ca^2+^]_i_ elevation in HEK293 cells [3]. Given the structural similarities and overlapping functions of Cx43 and Panx1, the physiological roles of the two proteins are still a subject of debate. Current methods to distinguish between the two channels rely on electrophysiological approaches (mainly single-channel conductance measurements) and peptide-based inhibitors targeting Cx43 or Panx1. However, single-channel conductance analyses are not without ambiguity, since Cx43 HCs show a fully open conductance in the order of 200–240 pS and a substate of ~80 pS, while Panx1 channel conductance has been reported to be as high as 550 pS in exogenously expressing Xenopus oocytes and between 25 and 100 pS in mammalian cells [4,5,6]. A second and commonly used strategy is to employ peptides designed to mimic specific regions in the native sequence of connexins and pannexins to interfere with channel function/gating. The mimetic peptides Gap19 and ^10^Panx1 are widely used inhibitory peptides for Cx43 and Panx1 channels. Gap19 is a nonapeptide mimicking a sequence of nine amino acids on the intracellular loop of Cx43 (KQIEIKKFK), which is crucial for the CT–CL interaction that is a prerequisite for HC gating [7,8]. The ^10^Panx1 peptide consists of a sequence from the first extracellular loop of Panx1 (WRQAAFVDSY), which is known to affect channel opening [9]. Previous research has shown that the extracellular application of 200 µM Gap19 does not affect ATP release from cells overexpressing Panx1 channels stimulated by high extracellular K^+^ depolarizing conditions, while the same concentration of ^10^Panx1 inhibits ATP release from these cells [10]. In another study, both ^10^Panx1 and Gap19 were tested in HeLa cells expressing Cx43 or Panx1 channels, providing evidence of good selectivity for both of these peptides at a lower extracellular concentration of 100 μM [11]. However, in these studies, the channel activity was investigated by indirect approaches, such as ATP release and dye uptake experiments, and a direct correlation between mimetic peptide concentrations and single-channel current events is still missing. It is worth noting that ^10^Panx1 has been reported to partially inhibit Cx46 HC peak currents [12,13], which further emphasizes the need to assess the selectivity of ^10^Panx1 and Gap19 for Panx1 and Cx43 channels by electrophysiological approaches. Distinguishing between Panx1- and Cx43 HC-related signaling pathways presents a major challenge; therefore, peptide inhibitors of the two channels are often used to determine which channel plays the dominant role. Here, we compared the effects of Gap19 and ^10^Panx1 on Cx43 and Panx1 channel activities to scrutinize the cross-selectivity of these peptides at concentrations that have been commonly employed. We recorded a large dataset of ~25,000 single-channel opening/closing events from Cx43- and Panx1-expressing mammalian cell lines and analyzed the effects of Gap19 and ^10^Panx1 on unitary gating transitions from Cx43 HC and Panx1.

## 2. Results

### 2.1. Panx1 Channel-Opening Activity Is Concentration-Dependently Inhibited by ^10^Panx1 Peptide While Gap19 Has No Effect below 500 µM

We used C6 cells stably transfected with Panx1 (C6-Panx1) [14] to determine the single-channel unitary current activity of Panx1 channels upon their exposure to Gap19 and ^10^Panx1 applied at concentrations of 100, 200 and 500 µM, which have been used in various studies, as summarized in Table 1.

The representative traces depicted in Figure 1A (enlarged display in Appendix A) show unitary Panx1 current activities evoked by repeated (every 40 s) V_m_ steps from a holding potential of −30 mV to +70 mV (with a duration of 30 s) in voltage-clamp experiments (using whole-cell patch-clamp recording). Under control conditions, a single channel conductance of ~65 pS was observed as determined from the channel transition histograms shown in Figure 1B, i.e., below the 100 pS range, which is typical for voltage-dependent Panx1 channel-opening activity [4,5,6]. We tested the effect of ^10^Panx1 and Gap19 on these currents. ^10^Panx1 was applied extracellularly while Gap19 was applied intracellularly (via a whole-cell recording pipet) as the target interaction sites of these peptides are located outside (^10^Panx1 is composed of a decapeptide sequence on the first extracellular loop of Panx1 [9], with some evidence pointing to an interaction with the pore of the channel protein [42]) and inside the cell (Gap19 is a mimetic peptide of a sequence on the intracellular loop of Cx43 that interacts with the Cx43 C-terminal tail [7,8,10]), respectively. 

Transition histograms report channel transition properties, such as single-channel conductance, but not dwell times in the open or closed states. Figure 1C depicts all-point histograms of the channel activity, showing the event counts in the closed and open states. The largest open-state peak had a ~71 pS single-channel conductance followed by a second smaller peak in the range of the double conductance level, indicating the simultaneous opening of two Panx1 channels. The ~71 pS was not significantly different from the ~65 pS value concluded from the transition analysis. The ^10^Panx1 peptide clearly decreased the frequency of open-state events, while Gap19 also seemed to have some effect at 200 µM. To better judge the effect of the two peptides, we further quantified the nominal open probability (NP_o_) of the Panx1 channels, as shown in Figure 2A. This analysis demonstrated that ^10^Panx1 significantly reduced the NP_o_ of the Panx1 channels by 26.0%, 49.5% and 66.0% for the concentrations of 100, 200 and 500 µM, respectively. In contrast, 100 and 200 µM Gap19 did not significantly affect the NP_o_ of the Panx1 channel; Gap19 did, however, significantly reduce the NP_o_ at 500 µM (a 26% inhibition; see Figure 2A). 

A careful inspection of the experimental traces of the Panx1 channel activity further demonstrated fast-flickering closures in the presence of ^10^Panx1, which were not observed with Gap19 (Appendix A). The flickering frequency increased with the concentration of ^10^Panx1 (67, 84 and 146 Hz for 100, 200 and 500 µM ^10^Panx1) and indicated fast ON–OFF gating effects within the channel pore [43].

### 2.2. Cx43 Hemichannel-Opening Activity Is Inhibited by Gap19 but Also by ^10^Panx1 at All Concentrations Tested

In the next step, we investigated the effects of the ^10^Panx1 peptide on Cx43 HC currents. Since the peptide is often used at higher concentrations to inhibit the Panx1 channel (Table 1), we investigated the effect of the ^10^Panx1 peptide on Cx43 HC currents over a wider concentration range (100, 200 and 500 μM). We used HeLa cells stably expressing Cx43 (HeLa-Cx43), which demonstrated typical Cx43 unitary current activity upon stepping from −30 mV to +70 mV (30 s; Figure 3, enlarged display in Appendix A) as previously reported [10]. Cx43 HC gating is characterized by a fully open state in the 200–240 pS range and a subconductance level in the order of 80 pS. In line with previous observations [16], Gap19 reduced the frequency of transitions to the fully open state and increased the propensity of subconductance transitions starting from 100 µM (Figure 3). Interestingly, ^10^Panx1 exerted a similar effect, increasing the frequency of subconductance transitions while decreasing the frequency of transitions to the fully open state. Note that the subconductance peaks in the transition histograms for the Gap19/^10^Panx1 peptide conditions look tall (Figure 3B), while the subconductance peaks are hardly visible in the all-point histograms (Figure 3C). This difference is the consequence of the short duration of substate openings (~170 ms on average) while the dwell times for the fully open state are much longer (up to a few seconds) with electrical stimulation as used here. As a result, the impact of increased subconductance gating on the total current flow as well as NP_o_ is limited. Overall, the NP_o_ of the Cx43 HC-opening activity clearly demonstrated a concentration-dependent and significant inhibition for Gap19 but also for ^10^Panx1 (Figure 2B), which reduced the NP_o_ of the Cx43 HCs by 37% at 100 µM, 83% at 200 µM and 94% at 500 µM (Figure 2B). 

## 3. Discussion

Among the connexin and pannexin families, Cx43 and Panx1 are the most commonly expressed isotypes in the human body. These pore-forming proteins present a strong similarity in terms of their membrane topology, channel gating, permeability profile, modes of activation and pathophysiological roles (as reviewed in [1,2,44]). As such, conclusions on the contribution of these channels in physiological or pathological responses have been largely based on the effect of peptide inhibitors. This study aimed to investigate the cross-effects of the Gap19 and ^10^Panx1 peptides on the currents associated with Cx43 HC and Panx1 that are activated by stepping to positive membrane potentials. Our data confirm that Gap19 concentration-dependently inhibits Cx43 HCs (Figure 2B) and acts as a gating modifier, reducing transitions to the fully open state (~210 pS) while increasing the number of transitions to the subconductance state (~80 pS, Figure 3). Gap19 had non-significant effects on Panx1 channel activity at intracellular concentrations up to 200 μM, while at 500 μM, the inhibitory effect on Panx1 was similar to that on 100 μM ^10^Panx1, causing a ~25% reduction in the NP_o_ (Figure 2A). 

The analysis of Figure 2 indicates that Gap19 applied intracellularly at 100 μM significantly reduces the NP_o_ of Cx43 HCs without affecting the Panx1 channels. Figure 2A shows that 200 µM extracellular ^10^Panx1, the concentration most frequently used in published studies (Table 1), brings about the half-maximal inhibition of the Panx1 channel’s NP_o_. However, this concentration decreases the NP_o_ of Cx43 HCs by ~83% (Figure 2B; for traces, see Figure 3). Surprisingly, ^10^Panx1 acted as a gating modifier of the Cx43 HCs at all the concentrations tested, in a similar fashion as previously reported for Gap19 [16], i.e., it reduced the propensity of gating to the main open state but increased the incidence of gating to the HC subconductance state (Figure 3). In contrast, ^10^Panx1 did not induce the subconductance gating of Panx1 channels (Figure 1). Previous work has suggested that the Gap19 disruption of the interaction between the CT and the CL may be involved in the increased substate gating of Cx43 HCs [16], so it remains to be determined whether ^10^Panx1 would have any such effects on CT–CL interactions. This brings us to the point of the membrane permeability of the Gap19 and ^10^Panx1 peptides. Sequence-based estimates of the cell-penetrating potential of candidate peptide sequences using the C2Pred webserver [45] indicated that ^10^Panx1 has poor membrane permeability. As such, the possibility that the peptide would make its way into the cell and interfere with CT–CL interactions is considered to be low. Possibly, ^10^Panx1 may indirectly affect CT–CL interactions via peptide interactions with the extracellular loops, thereby impacting the conformational state of the CT and CL structures inside the cell. In fact, such a scenario should also be considered for the Cx43 HC-inhibiting effects of Gap26 and Gap27 which contain amino acid sequences of extracellular loop 1 and 2 from the Cx43 protein.

In the present study, we applied Gap19 to the intracellular compartment via the whole-cell patch-clamp pipette. For comparison, the intracellular Gap19 concentrations approximately corresponded to the extracellular concentrations of membrane-permeable TAT-Gap19, which equilibrates at the two sides of the plasma membrane [10]. Consequently, the Gap19 concentrations used here approximately correspond to comparable concentrations of TAT-Gap19 applied extracellularly. Table 2 demonstrates that TAT-Gap19 concentrations of 100, 200 and 500 µM have been used in published experimental work.

In summary, the present data show that 100–200 μM intracellular Gap19 (equivalent to 100–200 µM TAT-Gap19 applied outside the cell) significantly inhibits Cx43 HCs without affecting Panx1 channel function. Contrastingly, extracellular ^10^Panx1 more strongly reduces Cx43 HC function than Panx1 channel function at all the concentrations tested, limiting the use of this peptide as a pharmacological tool to identify the involvement of Panx1 channels in cellular function.

To the best of our knowledge, this is the first report on the cross-effects of Gap19 and ^10^Panx1 based on electrophysiological experiments allowing for a direct readout of Cx43 HC and Panx1 channel function. Our analysis based on a large data set of channel activity provides increased insight into the gating behavior of Cx43 HCs and Panx1 channels. Establishing the gating profiles of these channels is intended to be a starting point for future studies comparing Cx43 HC and Panx1 channel function to identify their respective roles in cell signaling and to drive further experimental research to develop peptides and molecules with optimal target selectivity (as reviewed in [46]). Along this line, novel Panx1 inhibitors based on a quinoline or indole scaffold have been recently reported to display improved selectivity towards connexins [47,48].

**Table 2 ijms-24-11612-t002:** Concentrations used for extracellular application of TAT-Gap19.

Peptide	Concentration (μM)	Application	References
TAT-Gap19	100	extracellular	[49,50,51,52]
	200	extracellular	[10,24,53,54,55]
	500	extracellular	[24]

## 4. Materials and Methods

### 4.1. Chemicals and Reagents

Ethylene glycol-bis-(β-aminoethyl ether)-N,N,N’,N’-tetraacetic acid (EGTA), 5-tetraethylammonium (TEA)-Cl, pyruvic acid and 4-(2-hydroxyethyl)-1-piperazineethanesulfonic acid (HEPES) were purchased from Sigma-Aldrich (Bornem, Belgium). Gap19 (KQIEIKKFK) and ^10^Panx1 (WRQAAFVDSY) were synthesized to a purity of >90%.

### 4.2. Cell Cultures

We used HeLa cells stably transfected with Cx43 (HeLa-Cx43 [56]); endogenous expression of Panx1 is very low [57] or absent [58] in non-transfected HeLa cells. C6 glioma cells stably transfected with Panx1 (C6-Panx1, C-terminally *myc*-tagged; kind gift of Dr Christian C. Naus, University of British Columbia, Canada [14]) were used for the Panx1 channel studies; Cx43 expression is low [59,60] or undetectable [61] in these cells. HeLa cells were maintained in Dulbecco’s modified Eagle’s medium (DMEM; Invitrogen, Gent, Belgium), and C6 cells were grown in DMEM:Ham’s F12 (1:1—Invitrogen, Merelbeke, Belgium), all supplemented with 10% fetal bovine serum (FBS), 2 mM glutamine, 10 U/mL penicillin, 10 µg/mL streptomycin and 0.25 µg/mL fungizone. Stable exogenous expressions were maintained by supplementation of 1 and 3 µg/mL puromycin (Sigma-Aldrich, Bornem, Belgium) for HeLa-Cx43 and C6-Panx1 cells, respectively. Stable transfectants were subcultured into puromycin-free medium 2 days prior to experimentation. Cells were maintained at 37 °C and 10% (HeLa-Cx43) or 5% (C6-Panx1) CO_2_.

### 4.3. Electrophysiological Recordings

All experimental data were obtained with the patch-clamp technique in whole-cell recording mode on HeLa-Cx43 and C6-Panx1 cells. Extracellular solution was composed of the following (in mM): 150 NaCl, 4 CsCl, 2 CaCl_2_, 2 MgCl_2_, 2 pyruvic acid, 5 glucose and 5 HEPES at pH of 7.4, while the pipette solution consisted of the following (in mM): 125 CsCl, 10 sodium aspartate (NaAsp), 0.26 CaCl_2_, 1 MgCl_2_, 2 EGTA, 10 tetraethylammonium (TEA)-Cl and 5 HEPES at pH 7.2. The free Ca^2+^ concentration of the pipette solution was estimated to be ~50 nM as calculated with WEBMAX (https://somapp.ucdmc.ucdavis.edu/pharmacology/bers/maxchelator/webmaxc/webmaxcS.htm; accessed on 20 September 2020). Whole-cell recording was performed under control conditions and in the presence of Gap19 or ^10^Panx1. Gap19 was included in the pipette solution, while ^10^Panx1 was added externally and preincubated for 45 min. Recordings were commenced 2 min after establishment of the whole-cell configuration. Currents were recorded with an EPC 7 PLUS patch-clamp amplifier (HEKA Elektronik, Germany). Unitary current activities were elicited by stepping the cells from a holding potential of −30 mV to a membrane potential (V_m_) of +70 mV for 30 s. Data were digitized at 4 KHz using an NI USB-6221 data acquisition device (National Instruments, Austin, TX, USA) and WinWCP acquisition software designed by Dr. J. Dempster (University of Strathclyde, United Kingdom). Measured currents were filtered by a 7-pole Bessel low-pass filter at 1 KHz cut-off frequency. Transition analysis was performed as reported in [16]. Unitary channel conductances were determined from transition histograms as well as from all-point histograms and were obtained by fitting their frequencies to Gaussian distributions. Channel opening activity is expressed as nominal open probability NP_o_, denoting the number of active channels in the patch multiplied by the open probability. 

### 4.4. Statistical Analysis

Data were processed using Clampfit version 11.2 (Molecular Devices, San Jose, USA) and HemiGUI software as previously described [16]. Results are expressed as mean ± SEM (unless otherwise stated), with n giving the number of traces, ***N*** the number of cells, and N the number of experiments on different cell culture dishes. Multiple groups were compared by one-way analysis of variance with post-hoc Dunnett’s or Bonferroni multiple comparison tests, making use of GraphPad PRISM 5.0 (GraphPad Software, Inc., La Jolla, CA). Results were considered statistically significant when *p* ≤ 0.05 (*; # for *p* ≤ 0.05, **; ## for *p* ≤ 0.01, ***; ### for *p* ≤ 0.001, and ****; #### for *p* ≤ 0.0001).

## Figures and Tables

**Figure 1 ijms-24-11612-f001:**
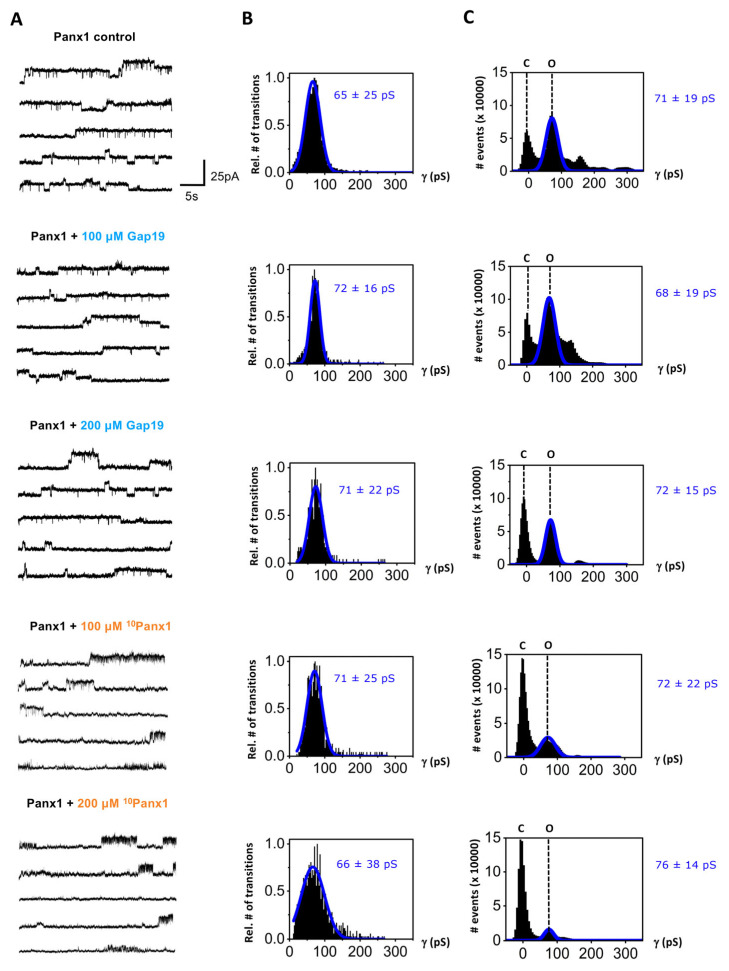
Effect of mimetic peptides Gap19 and ^10^Panx1 on current activity of Panx1 channels recorded in C6-Panx1 cells. Channel activity was evoked by stepping from a holding potential of −30 mV to +70 mV for 30 s. (**A**) Representative current traces demonstrating channel activity for control conditions and in the presence of Gap19 or ^10^Panx1. (**B**) Transition histograms demonstrating Gaussian distribution of channel-opening and -closing activity, with indication of the mean ± SD of unitary conductance of the recorded channel transition activity. The conductance values in the presence of the peptides were not significantly different from those under control. The number of transitions is expressed relative to the maximum observed in each of the experimental conditions shown (C6-Panx1 control N = 11, *N* = 30, n = 260; C6-Panx1 + 100 μM Gap19 N = 11, *N* = 12, n = 76; C6-Panx1 + 200 μM Gap19 N = 11, *N* = 9, n = 70; C6-Panx1 + 100 μM ^10^Panx1 N = 11, *N* = 11, n = 106; C6-Panx1 + 200 μM ^10^Panx1 N = 11, *N* = 12, n = 116). (**C**) All-point histograms of channel activity showing event counts in the closed (C) and open (O) states expressed as a function of channel conductance. In the presence of the peptide inhibitors, the open-state peaks decreased in height while the closed-state peaks increased, most clearly discernable for ^10^Panx1 but also to some extent for 200 µM Gap19; significances for these effects at the level of NP_o_ are given in Figure 2.

**Figure 2 ijms-24-11612-f002:**
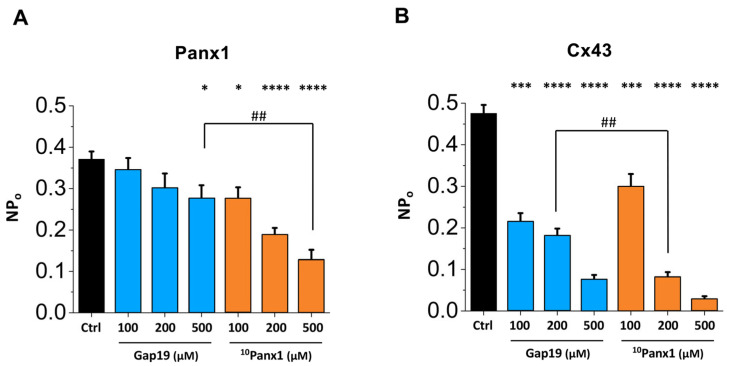
Summary graph of nominal open probability (NP_o_) data showing the effect of various Gap19 and ^10^Panx1 concentrations on Panx1 channels (**A**) and Cx43 HCs (**B**). Statistical analysis by one-way ANOVA with stars (*) indicating comparisons to the control condition (Ctrl) by Dunnett’s post-test and number signs (#) indicating significances between Gap19 and ^10^Panx1 with a Bonferroni post-test (one symbol for *p* ≤ 0.05, two symbols for *p* ≤ 0.01, three symbols for *p* ≤ 0.001 and four symbols for *p* ≤ 0.0001). (**A**) C6-Panx1 control = 0.37 ± 0.02 (N = 11, *N* = 30, n = 260); C6-Panx1 + 100 μM Gap19 = 0.34 ± 0.03 (N = 11, *N* = 12, n = 76); C6-Panx1 + 200 μM Gap19 = 0.30 ± 0.03 (N = 11, *N* = 9, n = 70); C6-Panx1 + 500 μM Gap19 = 0.28 ± 0.03 (N = 11, *N* = 10, n = 87); C6-Panx1 + 100 μM ^10^Panx1 = 0.28 ± 0.03 (N = 11, *N* = 11, n = 106); C6-Panx1 + 200 μM ^10^Panx1 = 0.19 ± 0.01 (N = 11, *N* = 12, n = 116); C6-Panx1 + 500 μM ^10^Panx1 = 0.13 ± 0.02 (N = 11, *N* = 11, n = 76). (B) HeLa-Cx43 control = 0.48 ± 0.02 (N = 11, *N* = 23, n = 190); HeLa-Cx43 + 100 μM Gap19 = 0.21 ± 0.02 (N = 11, *N* = 12, n = 98); HeLa-Cx43 + 200 μM Gap19 = 0.18 ± 0.02 (N = 11, *N* = 15, n = 122); HeLa-Cx43 + 500 μM Gap19 = 0.08 ± 0.01 (N = 11, *N* = 12, n = 88); HeLa-Cx43 + 100 μM ^10^Panx1 = 0.30 ± 0.03 (N = 11, *N* = 12, n = 88); HeLa-Cx43 + 200 μM ^10^Panx1 = 0.08 ± 0.01 (N = 11, *N* = 11, n = 97); HeLa-Cx43 + 500 μM ^10^Panx1 = 0.03 ± 0.01 (N = 11, *N* = 8, n = 53).

**Figure 3 ijms-24-11612-f003:**
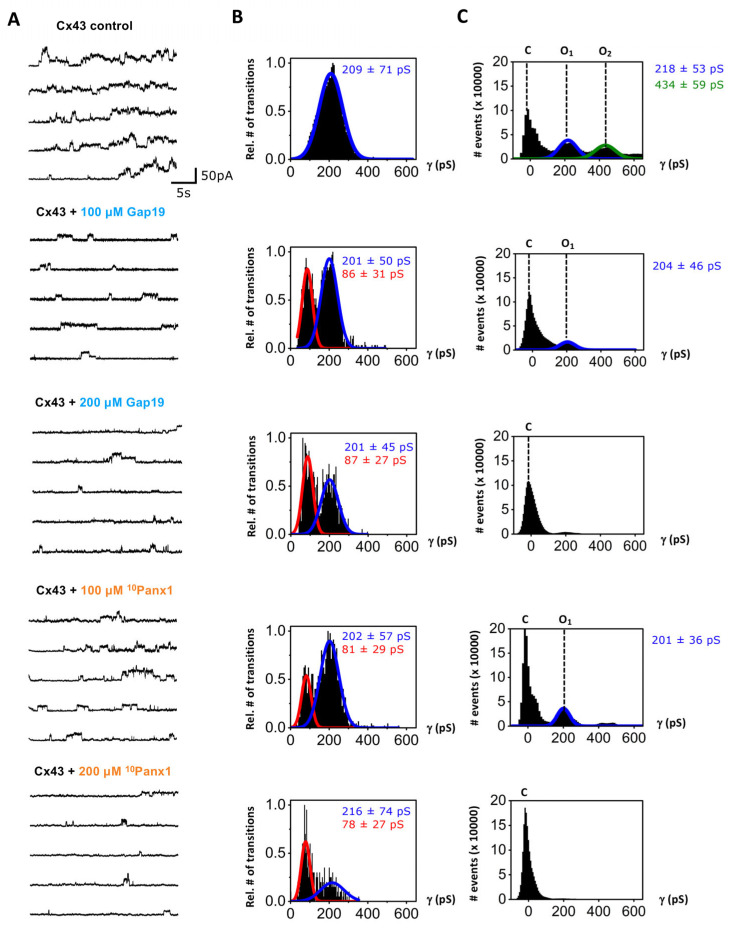
Effect of mimetic peptides Gap19 and ^10^Panx1 on HC current activity recorded in HeLa-Cx43 cells. Channel activity was evoked by stepping from a holding potential of −30 mV to +70 mV for 30 s. (**A**) Representative current traces demonstrating channel activity for control conditions and in the presence of Gap19 or ^10^Panx1. (**B**) Transition histograms demonstrating Gaussian distribution of channel opening and closing activity, with indication of the mean ± SD of the recorded channel transition activity. Gap19 as well as ^10^Panx1 increased gating to the subconductance state as indicated by the red marked peaks. The number of transitions was normalized to the maximum observed in each cluster of the experiment (HeLa-Cx43 control N = 11, *N* = 23, n = 190; HeLa-Cx43 + 100 μM Gap19 N = 11, *N* = 12, n = 98; HeLa-Cx43 + 200 μM Gap19 N = 11, *N* = 15, n = 122; HeLa-Cx43 + 100 μM ^10^Panx1 N = 11, *N* = 12, n = 88; HeLa-Cx43 + 200 μM ^10^Panx1 N = 11, *N* = 11, n = 97). (C) All-point histograms of channel activity showing event counts in the closed (C) and open states (O_1_ and O_2_) expressed as a function of channel conductance; O_2_ peaked at twice the O_1_ conductance, corresponding to stacked (simultaneous) opening of two Cx43 HCs.

**Table 1 ijms-24-11612-t001:** Gap19 and ^10^Panx1 concentrations that have been used to achieve Cx43 and Panx1 channel inhibition, respectively.

Peptide	Concentration (μM)	Application	References
Gap19	100	intracellular	[10,15,16,17]
	100	extracellular	[11,18,19,20,21]
	100	intracellular/extracellular	[22]
	200	intracellular	[23]
	200	extracellular	[10]
	400	intracellular	[10]
	400	extracellular	[24]
	500	extracellular	[10]
^10^Panx1	100	extracellular	[9,11,18,19,20,25,26,27,28,29]
	200	extracellular	[9,10,12,30,31,32,33,34,35,36,37]
	300	extracellular	[38]
	400	extracellular	[39,40]
	500	extracellular	[41]

## Data Availability

Not applicable.

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
