# Peer review of "Cx43 Hemichannel and Panx1 Channel Modulation by Gap19 and 10Panx1 Peptides"

_ijms, 2023, doi:10.3390/ijms241411612_

Round 1

Reviewer 1 Report

The specificity of “connexin mimetic” peptides as gap junction (GJ)/hemichannel (HC) blockers was questioned two decades ago by Gerhard Dahl, who claimed substantial inhibition of Panx1 by Cx43 EL peptides (see https://doi.org/10.1080/15419060801891018).  These studies have been largely ignored by investigators studying Cx43 HC (and are not cited in this manuscript). 

The authors compared effects on Panx1 transfected C6 glioma cells and Cx43 transfected HeLa cells of several concentrations of extracellularly applied 10Panx1 (which corresponds to a 10 AA neutral sequence in Panx1 EL1) and intracellularly applied Gap19 (corresponding to a 9 AA lysine rich peptide sequence in the Cx43 IL).  Membrane currents attributed to Panx1 or Cx43 HC channels (see comments below) were measured in whole cell configuration in response to steps from -30 to +70 mV.

Figure 2 compares blockade of Panx1 and Cx43 HC currents (NPo) by 0.1, 0.2 and 0.5 mM concentrations of each peptide.  Gap19 did not significantly affect Panx1 at 0.1 or 0.2 mM and at 0.5 mM reduced by 25%, whereas all concentrations of 10Panx1 decreased Panx1 NPo, maximally by 65% (Fig 2A).  Gap19 significantly affected Cx43 HC at all concentrations, (slightly more than 50% at 0.1 and 0.2 mM, maximally 83% at 0.5 mM; 10Panx1 also significantly affected Cx43 HC at all concentrations tested, being twice as effective as Gap19 on Cx43 HC at 0.2 mM and maximally reducing NPo by 96%).

While the authors are correct that 0.1 mM Gap19 reduced Cx43 HC more than the same concentration of 10Panx1 (l 19), this difference may or may not be significant (was ANOVA performed?), and at higher concentrations 10Panx1 was the more effective Cx43 HC blocker.  Nevertheless, the data in Fig 2A support the claim that although 0.2 mM Gap19 decreases Panx1 NPo by 19%, the reduction is not significant until it reaches 24% at 0.5 mM.

Concerns:

Conclusions depend on certainty that NPo is entirely recording activity of specific channels. Exactly which currents were measured in the Panx1 and Cx43 transfected cells; do the transfected cells totally lack the other channel?  L.94 states that ~65 pS channels were observed that were “in the below 100 pS range that is typical…” Were larger unitary events excluded from analysis? Were smaller events in the HeLa-Cx43 cells excluded? 

The reference for the stable HeLa-43 cells is #53, in which Cx43 was cut from a rat heart clone.  Is that the same sequence used in these studies? 

In the reference cited for the Panx1 transfectants (#13), rat Panx1 was C terminally tagged with EGFP or c-myc, and Panx1 transfection increased gap junction coupling, presumably through Cx43 channels.  This information should be provided.  How were the GJ HCs induced by Panx1 expression not problematic for your study?

Statistics: ANOVA is claimed; was this performed on data in Fig 2?

Representative current traces are too small to see carefully, with lots of white space separating the traces.  Suggest reducing number of traces to 3 for each treatment, and greatly reducing the space between them in Figs 1 and 3.

Previous studies claiming Panx1 blockade by “Cx mimetic” peptides should be referenced.

Author Response

Reviewer comment:

The specificity of “connexin mimetic” peptides as gap junction (GJ)/hemichannel (HC) blockers was questioned two decades ago by Gerhard Dahl, who claimed substantial inhibition of Panx1 by Cx43 EL peptides (see https://doi.org/10.1080 /15419060801891018). These studies have been largely ignored by investigators studying Cx43 HC (and are not cited in this manuscript).

Author response:

We agree with the reviewer and apologize for the paucity of references to Prof. Gerhard Dahl’s work. There is a reference to his work (reference 12 describing inhibition of connexin 46 by 10Panx1 Wang et al. 2007, PMID: 17652431) and now include the suggested Dahl 2007 reference (PMID: 18392993) on page 2 (l.68) of the revised ms. We also added two recent papers by Crocetti et al. 2021, 2023 from the Dahl group reporting on novel Panx-1 blockers with improved selectivity (PMID: 36768344, PMID: 34174741) at the end of the discussion (l.259).

Reviewer comment:

While the authors are correct that 0.1 mM Gap19 reduced Cx43 HC more than the same concentration of 10Panx1 (l 19), this difference may or may not be significant (was ANOVA performed?), and at higher concentrations 10Panx1 was the more effective Cx43 HC blocker.

Author response:

Thanks for the comment. We checked this and indeed found that some of the comparisons between Gap19 and 10Panx1 are significant. In particular, 500 µM 10Panx1 more strongly inhibited the NPo of Panx1 channels compared to 500 µM Gap19 (Figure 2A). Moreover, 200 µM 10Panx1 more strongly inhibited the NPo of Cx43 hemichannels compared to 200 µM Gap19 (Figure 2A). These significant differences have now been added to Figure 2 in the revised version.

Reviewer comment:

Exactly which currents were measured in the Panx1 and Cx43 transfected cells; do the transfected cells totally lack the other channel?

Author response:

Endogenous expression of Panx1 is very low (PMID: 17009242) or absent (PMID: 16690210) in Hela cells, as demonstrated previously. Likewise, C6 cells have been shown to either exhibit low (PMID: 1650934; PMID: 1848013) or undetectable levels of Cx43 (PMID: 23095853). We have added this information in the M&M of the revised paper (l.271-275).

Reviewer comment:

Were larger unitary events excluded from analysis? Were smaller events in the HeLa-Cx43 cells excluded?

Author response:

The automated analysis procedure does not involve any filtering or thresholding of the traces analyzed (PMID: 33027889). The maximum of the histogram conductance axis of Panx1 channels and Cx43 hemichannels was determined by the inclusion of simultaneous, so-called ‘stacked’ opening of two channels.

Reviewer comment:

The reference for the stable HeLa-43 cells is #53, in which Cx43 was cut from a rat heart clone. Is that the same sequence used in these studies?

Author response:

Yes, this is indeed the case.

Reviewer comment:

In the reference cited for the Panx1 transfectants (#13), rat Panx1 was C terminally tagged with EGFP or c-myc, and Panx1 transfection increased gap junction coupling, presumably through Cx43 channels. This information should be provided. How were the GJ HCs induced by Panx1 expression not problematic for your study?

Author response:

We thank the reviewer for this remark and apologize for absence of the myc tag information, which is now added in l.273-274 of the revised paper.

Cx43 expression in C6 cells is low or undetectable (PMID: 1650934; PMID: 1848013; PMID: 23095853), as already referred to higher. Lai et al. 2007 suggested that Panx1 might form functional intercellular channels when stably expressed in C6 cells, and Palacios-Prado et al. 2022 recently reported on cell–cell channels formed by Panx1 (PMID: 35486697). Given the fact that all patch-clamp work was done on single cells, gap junctions or intercellular channels is not an issue here.

Reviewer comment:

Statistics: ANOVA is claimed; was this performed on data in Fig 2?

Author response:

Thank you for the comment. Indeed, in Fig. 2 multiple groups were compared by 1-way ANOVA with post-hoc Dunnett's multiple comparison test. This information has been added to the figure legend on l.143-144.

Reviewer comment:

Representative current traces are too small to see carefully, with lots of white space separating the traces. Suggest reducing number of traces to 3 for each treatment, and greatly reducing the space between them in Figs 1 and 3.

Author response:

We appreciate your point, however, traces displayed in Figs. 1 and 3 are scaled according to control conditions where channel activity is high due to multiple stacked openings. We now show in Figure S1 the enlarged display of representative current traces in Figure 1A, and in Figure S3 the enlarged display of representative current traces in Figure 3A.

Reviewer comment:

Previous studies claiming Panx1 blockade by “Cx mimetic” peptides should be referenced.

Author response:

We thank the reviewer for this remark and we added the paper by Dahl 2007 in l.68 as mentioned above.

Reviewer 2 Report

The paper is clearly written, easy to follow and "to the point".

Authors present evidence for the following statements:

Panx1 channel opening activity is inhibited by 10Panx1 peptide, but Gap19 has no effect below 500 μM.

Cx43 hemichannel opening activity is inhibited by Gap19 but also by 10Panx1 at several concentrations, which poses a limitation to the Panx1 selectivity of the 10Panx1 peptide.

Authors also discuss, the role of Gap19, being a gating modifier, that reduces the transitions to the fully open state, while increasing the number of transitions to the subconductance state.

As peptide selectivity is still an issue between Cx HCs and pannexins, this paper is important, because it gives direct electrophysical evidence of how these two inhibitors exert their effects.

I have only one question:

It would be interesting to read in the discussion, whether Gap26 or Gap27 modulate Cx43 in a similar manner as Gap19. This would be interesting, as they originate from the extracellular loops of Cx43HC, similarly to 10Panx1 which originates from the extracellular part of Panx1.

Author Response

Reviewer comment:

It would be interesting to read in the discussion, whether Gap26 or Gap27 modulate Cx43 in a similar manner as Gap19. This would be interesting, as they originate from the extracellular loops of Cx43HC, similarly to 10Panx1 which originates from the extracellular part of Panx1.

Author response:

We thank the reviewer for this interesting and plausible suggestion that Gap26/27 peptide interaction with the extracellular loops of Cx43 may also affect conformation of the C-terminal tail. We have added this in the discussion now mentioning: “In fact, such scenario should also be considered for the Cx43 HC-inhibiting effects of Gap26 and Gap27 that contain amino acid sequences of extracellular loop 1 and 2 from the Cx43 protein.” (l.234-236).